# I$_h$ block reveals separation of timescales in pyloric rhythm response to temperature changes in *Cancer borealis*

**Kyra Schapiro[1]\*, JD Rittenberg[1], Max Kenngott[1], Eve Marder[1,2]**

[1]Biology Department, Brandeis University, Waltham, United States; [2]Volen Center and Biology Department, Brandeis University, Waltham, United States

**Abstract** Motor systems operate over a range of frequencies and relative timing (phase). We studied the role of the hyperpolarization-activated inward current (I$_h$) in regulating these features in the pyloric rhythm of the stomatogastric ganglion (STG) of the crab, *Cancer borealis,* as temperature was altered from 11°C to 21°C. Under control conditions, rhythm frequency increased monotonically with temperature, while the phases of the pyloric dilator (PD), lateral pyloric (LP), and pyloric (PY) neurons remained constant. Blocking I$_h$ with cesium (Cs$^+$) phase advanced PD offset, LP onset, and LP offset at 11°C, and the latter two further advanced as temperature increased. In Cs$^+$ the frequency increase with temperature diminished and the Q$_{10}$ of the frequency dropped from ~1.75 to ~1.35. Unexpectedly in Cs$^+$, the frequency dynamics became non-monotonic during temperature transitions; frequency initially dropped as temperature increased, then rose once temperature stabilized, creating a characteristic 'jag'. Interestingly, these jags persisted during temperature transitions in Cs$^+$ when the pacemaker was isolated by picrotoxin, although the temperature-induced change in frequency recovered to control levels. Overall, these data suggest that I$_h$ plays an important role in maintaining smooth transitory responses and persistent frequency increases by different mechanisms in the pyloric circuitry during temperature fluctuations.

**\*For correspondence:**
kaschapiro@aol.com

**Competing interest:** The authors declare that no competing interests exist.

## eLife assessment

This **important** study investigates neurobiological mechanisms underlying the maintenance of stable, functionally appropriate rhythmic motor patterns during changing environmental conditions - temperature in this study in the crab *Cancer borealis* stomatogastric central neural pattern generating circuits producing the rhythmic pyloric motor pattern, which is naturally subjected to temperature perturbations over a substantial range. The authors present **compelling** evidence that the neuronal hyperpolarization-activated inward current (Ih), known to contribute to rhythm control, plays a vital role in the ability of these circuits to appropriately adjust the frequency of rhythmic neural activity in a smooth monotonic fashion while maintaining the relative timing of different phases of the activity pattern that determines proper functional motor coordination transiently and persistently to temperature perturbations. This study will be of interest to neurobiologists studying rhythmic motor circuits and systems and their physiological adaptations.

## Introduction

Many repetitive motor patterns, such as breathing, walking, and swimming, operate over a wide range of frequency and relative timing of component movements (phase). The frequency and phase of these components dictates the function and efficiency of the overall motion. Many locomotor systems maintain effectiveness by changing phase concomitantly with frequency (*Grillner, 1981*).

Some systems, such as lamprey (*Cohen et al., 1992*) and leech (*Marder and Calabrese, 1996*) swimming, require constant phase across frequencies. In either case, when such systems transition to new steady frequencies, they normally do so in a smooth, monotonic fashion. However, control theory recognizes that while smooth transitions may be optimal, they are not inevitable. The constraints of a system can cause a non-monotonic response to a state change, in some cases even having an initial change opposite to the final appropriate response. While the dynamics of many motor patterns under stable conditions have been extensively studied, how such systems appropriately adjust their phase and frequency as conditions change and avoid turbulent dynamics has been less often addressed.

Many rhythmic neurons and muscles utilize the hyperpolarization-activated inward current, $I_h$, in generating their rhythmic activity (*Robinson and Siegelbaum, 2003*; *Biel et al., 2009*; *DiFrancesco, 1985*). $I_h$ is mediated by the HCN-family of channels and is activated by hyperpolarization. This property allows excitable cells to depolarize after periods of inhibition, contributing to post-inhibitory rebound bursting behavior (*Robinson and Siegelbaum, 2003*; *Biel et al., 2009*; *McCormick and Pape, 1990b*; *McCormick and Pape, 1990a*; *Goaillard et al., 2010*). While some systems produce patterns myogenically, e.g., the vertebrate heartbeat (*DiFrancesco, 1985*; *Hagiwara and Irisawa, 1989*; *Tsien et al., 1972*; *DiFrancesco and Tromba, 1987*; *Yanagihara and Irisawa, 1980*), other systems rely on networks of cells to produce their activity, such as the leech heartbeat generator (*Nadim et al., 1995*; *Angstadt and Calabrese, 1989*; *Calabrese et al., 1995*; *Wenning et al., 2004*; *Stent et al., 1979*; *Tobin and Calabrese, 2005*) or the mammalian respiratory system (*Thoby-Brisson et al., 2000*; *Thoby-Brisson et al., 2003*).

While $I_h$ is utilized for rhythmogenesis in these systems, even a basic pattern of activity of a hard-wired neural circuit can be achieved through multiple combinations of intrinsic and synaptic properties (*Alonso and Marder, 2020*; *Alonso and Marder, 2019*; *Schulz et al., 2006*; *Schulz et al., 2007*; *Prinz et al., 2004*; *Golowasch et al., 1999a*; *Golowasch et al., 1999b*). The set of properties that allow functional activity during one set of conditions is likely inadequate for all conditions. Theoretical studies suggest that parameter sets that can produce similar activity in one environment display different robustness when the environment changes (*Alonso and Marder, 2020*; *Alonso and Marder, 2019*). As $I_h$ is a key component of ubiquitous rhythmic activity, we explored its potentially changing role during transitions in conditions a rhythmic system might encounter.

To this end, we utilized the response of the pyloric rhythm of the stomatogastric ganglion (STG) of the crab, *Cancer borealis*, to temperature changes (*Alonso and Marder, 2020*; *Marder and Bucher, 2007*; *Tang et al., 2010*; *Rinberg et al., 2013*). The pyloric rhythm controls food filtering by the crustacean pylorus (*McGaw and Curtis, 2013*). Frequency is dictated by a three neuron electrically coupled pacemaker kernel, made up of the intrinsically oscillating anterior burster (AB) neuron and the two pyloric dilator (PD) neurons. The pacemaker kernel neurons make inhibitory synapses onto the lateral pyloric (LP) neuron and several pyloric (PY) neurons, whose properties are such that LP recovers from pacemaker inhibition before PY (*Marder and Bucher, 2007*; *Hartline, 1979*; *Rabbah and Nadim, 2005*; *Hartline and Gassie, 1979*). At ~11°C the frequency of the pyloric rhythm is typically about 1 Hz with a characteristic triphasic rhythm of activity in the PD, LP, and PY neurons (*Hamood et al., 2015*).

As a poikilotherm, temperature changes are relevant to a crab in its natural environment. The pyloric rhythm frequency in *C. borealis* increases as temperature increases (*Alonso and Marder, 2020*; *Tang et al., 2010*; *Rinberg et al., 2013*; *Haddad and Marder, 2018*; *Soofi et al., 2014*), but the phasing of PD, LP, and PY neurons remains constant (*Tang et al., 2010*; *Soofi et al., 2014*). Previous work has demonstrated that $I_h$ increases in pyloric cells as temperature increases, but not whether this increase is necessary for normal responses of pyloric activity to temperature (*Tang et al., 2010*; *Peck et al., 2006*). To probe the role $I_h$ plays across changing conditions, we pharmacologically blocked $I_h$ using cesium ions during temperature changes and observed the effect of this block on the frequency increase and phase constancy of the pyloric rhythm. This revealed unanticipated changes in frequency during acute transitions between temperatures.

## Results

### The steady-state frequency increase in response to elevated temperature was decreased by $Cs^+$

We used a combination of extracellular and intracellular recordings to monitor pyloric activity while the STG was superfused sequentially with saline or saline with 5 mM CsCl to block $I_h$ (*Peck et al., 2006*; *Harris-Warrick et al., 1995*; *Goeritz et al., 2011*; *Ballo et al., 2010*; *Zhang et al., 2003*; *Zhu et al., 2016*). We began recording at the standard temperature of 11°C and changed the temperature of the superfused solution in steps of 2°C up to 21°C using a Peltier device. The dish temperature and pyloric activity were allowed to stabilize for 4 min between steps (*Figure 1A*). To confirm the effectiveness of $Cs^+$ in reducing $I_h$ across this temperature range, we voltage clamped pyloric neurons (PD and LP) at the temperature extremes used in the experiment, 11°C and 21°C (*Figure 1—figure supplement 1*). We held cells at –50 mV and hyperpolarized for 12 s in –10 mV steps from –100 mV to –120 mV. Because $I_h$ is slow to activate, we quantified $I_h$ as the difference between the initial current needed to hold the hyperpolarization and the current required to hold the hyperpolarization at steady state. At 11°C and –110 mV, $I_h$ was 1.5±0.83 nA in saline and 0.23±.5 nA in $Cs^+$, a reduction of 93±30% (4 PD cells, 3 LP cells, paired Student's t-test; p=0.0005). At 21°C and –110 mV, $I_h$ was 3.2±2.3 nA in saline and 0.6±.8 nA in $Cs^+$, a reduction of 78±20% (5 LP cells, 3 PD cells, paired Student's t-test; p=0.02).

An example of simultaneous intracellular PD and LP activity during both control saline and in $Cs^+$ at each temperature is shown in *Figure 1B*. As previously reported (*Tang et al., 2010*; *Rinberg et al., 2013*; *Soofi et al., 2014*), the pyloric frequency increased as temperature increased in control (*Figure 1B*, top traces). In $Cs^+$ at 11°C, the pyloric frequency was significantly decreased compared to control conditions (saline: 1.2±0.2 Hz; $Cs^+$ 0.9±0.2 Hz, paired Student's t-test; p<0.001) as has also been previously reported (*Peck et al., 2006*; *Zhang et al., 2003*; *Zhu et al., 2016*; *Soofi and Prinz, 2015*).

Interestingly, the degree of frequency increase with temperature was diminished in $Cs^+$ (*Figure 1B*, bottom traces). To quantify this effect, we took the mean pyloric frequency during times when the preparation was within 0.3°C of the target temperature and plotted the mean frequency as a function of temperature (*Figure 1C*). Within a preparation, the difference in frequency between control and $Cs^+$ at a given temperature significantly increased as temperature increased (*Figure 1D*). Additionally, we characterized the temperature sensitivity of each preparation in control and $Cs^+$ using $Q_{10}$ as a measure of frequency change with temperature as described previously (*Tang et al., 2010*). The mean $Q_{10}$ of the pyloric frequency was significantly lower in $Cs^+$ (1.3±0.5) than in control (1.7±0.3) (*Figure 1E*). Together, these analyses show that the temperature-induced increase in pyloric frequency in control saline was greater than in $Cs^+$.

In addition to counterbalancing the starting condition, to further support the conclusion that the decline in pyloric frequency $Q_{10}$ in $Cs^+$ was due to $Cs^+$ and not consecutive temperature changes, we performed two consecutive temperature changes in saline. The mean difference in frequency between 21°C and 11°C for the first (0.9±0.3 Hz) and second temperature changes (0.8±0.3 Hz) were not significantly different (*Figure 1F*). Moreover, the mean $Q_{10}$ of the first temperature increase (2.0±0.4) was not significantly different than the mean $Q_{10}$ of the second temperature increase (1.8±0.4) (*Figure 1E*). A two-sample t-test did not find a significant difference between the mean frequency $Q_{10}$ in saline in experiments that underwent two saline temperature changes compared to those that experienced one in saline and one in $Cs^+$ (*Figure 1E*). Therefore, the decrease in frequency-temperature sensitivity in $Cs^+$ was likely not a product of sequential temperature changes.

### $Cs^+$ revealed dynamic transitory frequency response: frequency decreased while temperature increased, then increased when temperature stabilized

While the increase in frequency over the temperature steps was primarily monotonic in control, it was often non-monotonic in $Cs^+$. In $Cs^+$, during the increases between temperatures there was often either no increase or a decrease in frequency. *Figure 2* illustrates simultaneous intracellular recordings of a PD cell and an LP cell during the beginning of the step from 13°C to 15°C in both control (*Figure 2A*) and $Cs^+$ (*Figure 2B*). In control, when the temperature was increased (maroon vertical dashed line),

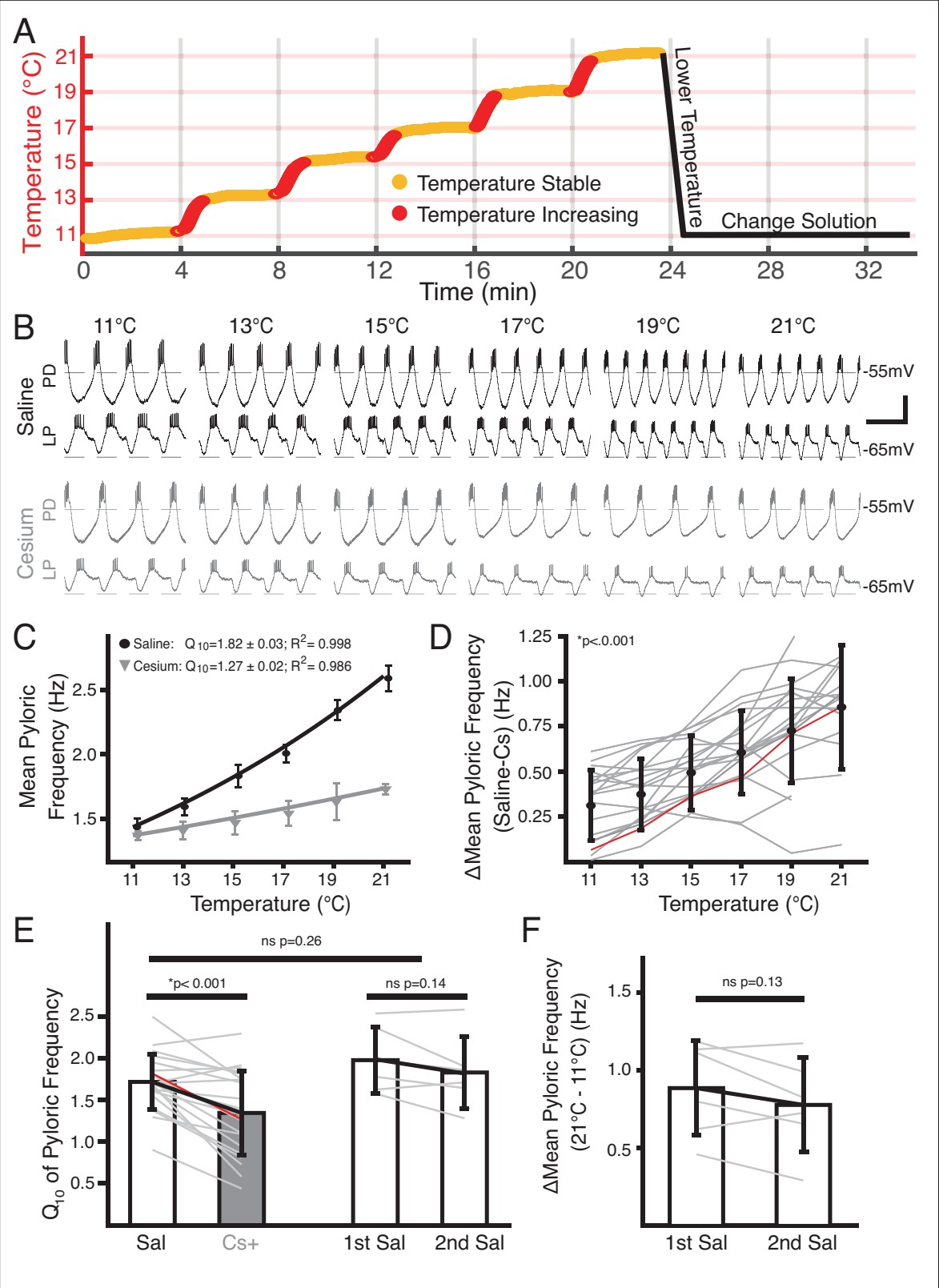

**Figure 1.** Temperature-induced frequency increase was diminished by Cs+ application. (**A**) Basic experimental protocol. Experiment began in saline or Cs+ solution (order randomized) at 11°C. The superfusion solution temperature was changed in steps of 2°C and allowed to stabilize over 4 min. Red represents times when the temperature was increasing, yellow when the temperature was considered stable. After the holding period at 21°C the preparation was returned to 11°C. The solution was then changed to the remaining condition over 10 min and the step protocol was repeated. (**B**)

*Figure 1 continued on next page*

*Figure 1 continued*

Intracellular recordings from a single preparation while in saline (top) and Cs⁺ (bottom) at each holding temperature (yellow in **A**) from 11°C to 21°C. Within a condition the top trace is pyloric dilator (PD) and the bottom trace is lateral pyloric (LP). Burst frequency increased as temperature increased in saline, but less so in Cs⁺. Scalebar: horizontal = 1 s, vertical = 20 mV. (**C**) Mean frequency ± SD at each holding temperature across conditions from a different preparation than in **B**. Line represents the best fit $Q_{10}$ equations for each condition; preparation in saline had a larger $Q_{10}$ than in Cs⁺ and as such a larger frequency increase with temperature. (**D**) Mean ± SD difference in frequency between saline and Cs⁺ across temperatures and preparations. As temperature increased, the difference between a preparation's frequency in saline and Cs⁺ significantly increased (one-way repeated measures ANOVA [$F(5,85) = 43.259$, $p<0.001$, N=21]). Gray lines represent individual experiments, red line represents experiment depicted in (**C**). (**E**) $Q_{10}$ was significantly lower in Cs⁺ than in saline (paired Student's t-test, $p<0.001$, N=21). $Q_{10}$ was not significantly different in two consecutive saline temperature changes (paired Student's t-test, $p=0.14$, N=6). $Q_{10}$ was also not significantly different between the mean $Q_{10}$ of the experiments with two saline temperature changes those that experienced one saline and one Cs⁺ temperature change (two-sample t-test, $p=0.26$). Bars are mean ± SD, individual lines represent individual experiments, red line represents experiment depicted in **C**. (**F**) Overall change in mean pyloric frequency from 11°C to 21°C was not significantly different between first and second temperature change in saline (paired Student's t-test, $p=0.14$, N=6).

The online version of this article includes the following figure supplement(s) for figure 1:

**Figure supplement 1.** Voltage clamp traces of pyloric dilator (PD) and lateral pyloric (LP) cells at the temperature extrema of these experiments in both saline and Cs⁺.

there was a net increase in frequency. In contrast in Cs⁺, there was an immediate decrease in frequency. In this particular example, the number of spikes per burst in LP decreased, though this effect was not seen in all preparations. Alignment of each PD cycle during this window to the first PD spike reveals that the decrease in frequency was due to elongation of the depolarizing phase and not the bursting, repolarization, or inhibitory phases, rather than a uniform expansion of all waveform components (*Figure 2C*).

*Figure 3A* shows an example of the entire frequency response during a temperature step between 17°C and 19°C. In control (*Figure 3A*, left), the frequency rose when the temperature increased, and stabilized when the temperature stabilized. Conversely in Cs⁺, the frequency decreased when the temperature increased as in *Figure 2B*, and then rose when the preparation reached the holding temperature (*Figure 3A*, right). This activity produced a 'jag' in the frequency-temperature curve (*Figure 3B*). We compared the distributions of change in mean frequency at times when the temperature was increasing or stable across conditions. 'Increasing temperature' was defined as times when the change in mean temperature was greater than 0.01°C (red on *Figures 1A, 3A and B*). For these increasing-temperature pyloric cycles we calculated the proportion in which the mean frequency decreased by more than 0.002 Hz (blue points in *Figure 3A*; left of dashed line in overall distribution for one experiment in *Figure 3C* top) in both control and Cs⁺. The proportion of cycles where frequency decreased while temperature increased was greater in Cs⁺ than in control for 20 of 21 experiments. In Cs⁺, the mean percent of temperature-increasing pyloric cycles in which the frequency decreased (31.4±15%) was significantly higher than in control (7.5±5%) (*Figure 3D*, paired Student's t-test, N=21, $p<0.001$). During pyloric cycles with temperature stability (yellow on *Figures 1A, 3A and B*), we calculated the proportion in which the mean frequency increased by more than 0.002 Hz (green points in *Figure 3A*; right of dashed line in overall distribution for one experiment in *Figure 3C* bottom). Again, this proportion was larger in Cs⁺ than in control in 20 of 21 experiments. In Cs⁺, the mean percent of temperature-stable cycles in which frequency increased in Cs⁺ (27.6±6%) was significantly higher than in control (14.1±5%) (*Figure 3E*, paired Student's t-test, N=21, $p<0.001$). To summarize, in Cs⁺ there was a distinct tendency for the frequency to decrease when temperature increased and then increase after the temperature stabilized.

## In Cs⁺, follower neurons determined frequency-temperature sensitivity: pacemaker kernel determined jag response

We investigated to what extent the two effects of Cs⁺, change in temperature sensitivity and non-monotonic frequency response, were caused by effects in the pacemaker kernel (AB and PD neurons) vs the follower neurons (LP and PY). To do so, we repeated the temperature step protocols in control saline, 5 mM Cs⁺ saline, and 5 mM Cs⁺+10⁻⁵ picrotoxin (PTX) saline. PTX blocks the glutamatergic synapse that is the only feedback from the follower neurons to the pacemaker (*Rinberg et al., 2013*; *Eisen and Marder, 1984*; *Eisen and Marder, 1982*; *Bidaut, 1980*; *Figure 4A*). *Figure 4B* illustrates that in Cs⁺+PTX, the pyloric rhythm did not show the diminished

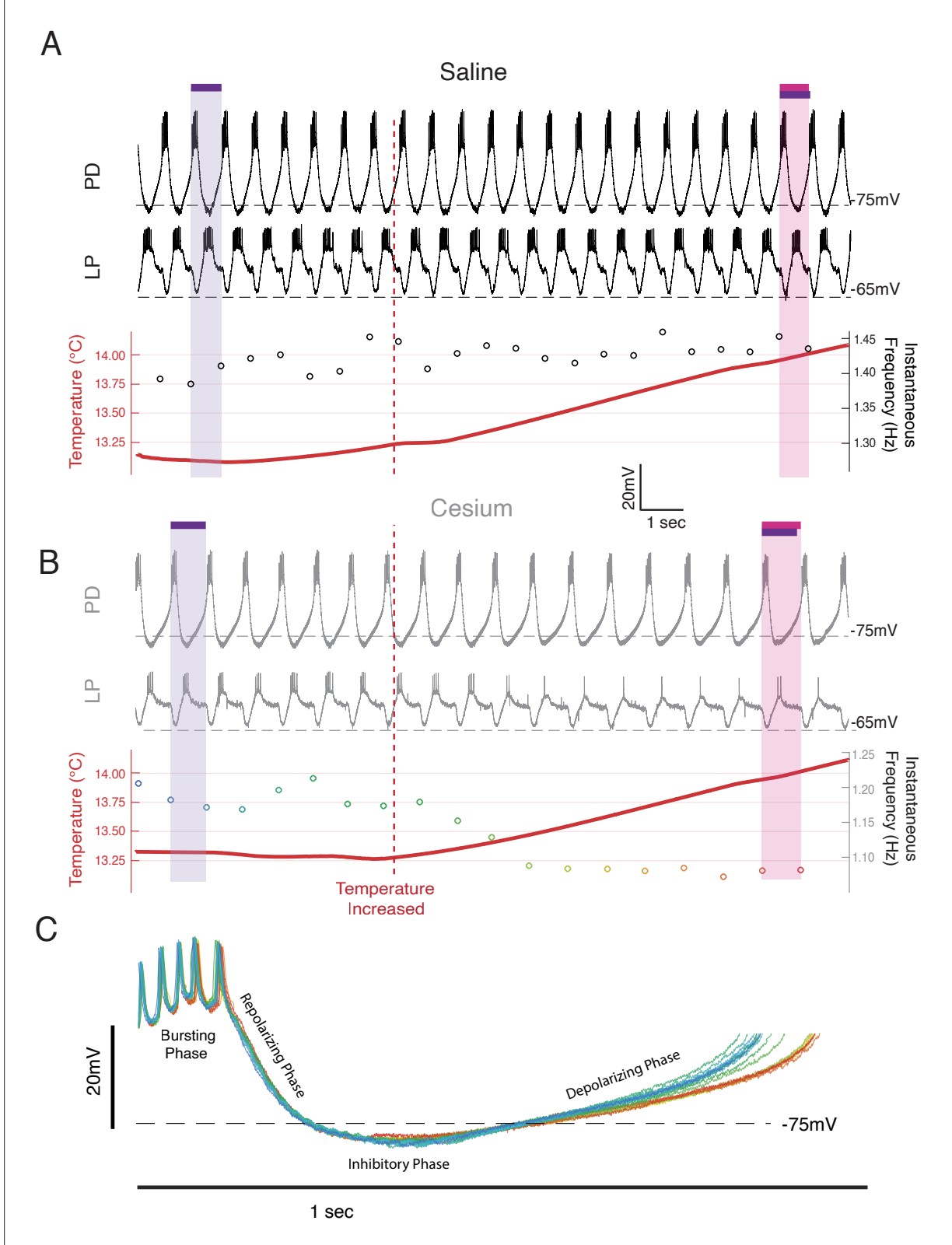

**Figure 2.** Frequency decreased when temperature was increased in Cs⁺ through an elongation of the depolarizing phase. Example from a single preparation as temperature was changed from 13°C toward 15°C in saline (**A**) and Cs⁺ (**B, C**). (**A, B**) Top: the intracellular recording from pyloric dilator (PD); Middle: simultaneous intracellular recording from lateral pyloric (LP); Bottom: simultaneous temperature (red; left axis) and instantaneous frequency (right axis). Purple bar highlights one pyloric period before the temperature was increased, while the magenta bar highlights one pyloric

*Figure 2 continued*

period after. (**A**) In saline when the temperature increased (maroon line), the period decreased as highlighted by the magenta bar being shorter than the purple bar. (**B**) In Cs$^+$, the period increased after the temperature shift, as highlighted by the magenta bar being longer than the purple bar. The same scale is used in both **A** and **B**. (**C**) Individual PD cycles from **B** aligned to the first spike. Coloration indicates when an individual cycle occurred as keyed by the instantaneous frequency trace in **B**. Increased period corresponds to an elongation of the depolarizing phase and a maintenance of the bursting and repolarization phases.

frequency increase seen in Cs$^+$ alone but did demonstrate jags. We compared the change in frequency between 11°C and 21°C in control, Cs$^+$, and Cs$^+$+PTX and found that there was a significant effect of condition; change in frequency in Cs$^+$ was significantly lower than in either control or Cs$^+$+PTX (*Figure 4C*) with no significant difference between control and Cs$^+$+PTX. Therefore, the effect of Cs$^+$ on the temperature-induced frequency increase appears to require the activity of the follower neurons.

We also compared the probability of frequency decrease during temperature increase and probability of frequency increase during temperature stability. Both measures showed significant effects of condition. The probability of frequency decrease while temperature increased was lower in control than in either Cs$^+$ or Cs$^+$+PTX (*Figure 4D*), with no difference between Cs$^+$ and Cs$^+$+PTX. Interestingly, the probability of frequency increase while temperature was stable was significantly different between all three conditions: control was lowest, Cs$^+$ was in the middle, and Cs$^+$+PTX was greatest. Therefore, the frequency jags appear to result from Cs$^+$ effects on the pacemaker kernel neurons, not the follower neurons, as they were still present when the isolated pacemaker underwent temperature steps in Cs$^+$.

One Cs$^+$+PTX experiment in which elevating the temperature produced a particularly pronounced transient decrease in frequency is shown in *Figure 4F*. Overlay of the PD cycles during this window aligned to the first spike again show that the decrease in frequency is due to an elongation of the depolarizing phase rather than a uniform expansion of the waveform (*Figure 4G*). Additionally, without the inhibition from the LP to PD synapse, the hyperpolarization associated with increased temperature (*Morozova et al., 2022*) becomes more apparent. When in this experiment the frequency increased after temperature had stabilized, the repolarizing phase shortened relative to the pre-temperature change, consistent with an overall maintenance of phase as temperature changed.

## Phase of PD and LP advanced in Cs$^+$

We also examined the effect of Cs$^+$ on the firing time and phase relationships of PD, LP, and PY. *Figure 5A* illustrates the pyloric activity in both control (left) and Cs$^+$ (right) at 11°C recorded on the pyloric dilator nerve (*pdn*) (top), lateral ventricular nerve (*lvn*) (middle), and pyloric constrictor nerve (*pyn*) (bottom). The time of PD bursting was significantly but modestly shorter in Cs$^+$ (*Figure 5B*, saline: 164±34 ms, Cs$^+$: 144±18 ms). Phase was quantified in *Figure 5C* and *Table 1*. Given the increase in period, it follows that the phase of PD OFF (percentage of period at which PD stops firing) was significantly advanced in Cs$^+$ compared to control. LP onset, which follows PD offset, was also phase advanced in Cs$^+$ conditions compared to control. However, this phase advance was more than would be expected given respective periods and the advanced PD OFF phase, as the delay between PD OFF and LP ON phase was significantly smaller in Cs$^+$ compared to control (*Figure 5D*, saline: 0.21±0.06, Cs$^+$: 0.17±0.04).

LP OFF was also significantly advanced in Cs$^+$, and subsequently duty cycle (percent of the period a neuron is firing) was preserved (paired Student's t-test, N=21, p=0.448). PY ON and PY OFF were not significantly different between Cs$^+$ and control. The intracellular waveforms were similar between Cs$^+$ and control conditions apart from the noted phase shifts. This similarity is illustrated in *Figure 5E* where two cycles of each pyloric neuron's intracellular activity have been scaled to the same period.

We identified cycles where period was matched between control at 11°C and Cs$^+$ (typically at a higher temperature for Cs$^+$). Although PD burst duration was only 51±30 ms shorter in Cs$^+$, LP fired 91±68 ms earlier. Conversely, PY latency did not change in these matched frequency samples (1.9±100 ms earlier). There was no correlation between how advanced PD's offset was between control and Cs$^+$ conditions and how much LP's onset advanced in these matched period samples (Pearson correlation; r(21)=0.26, p=0.2928).

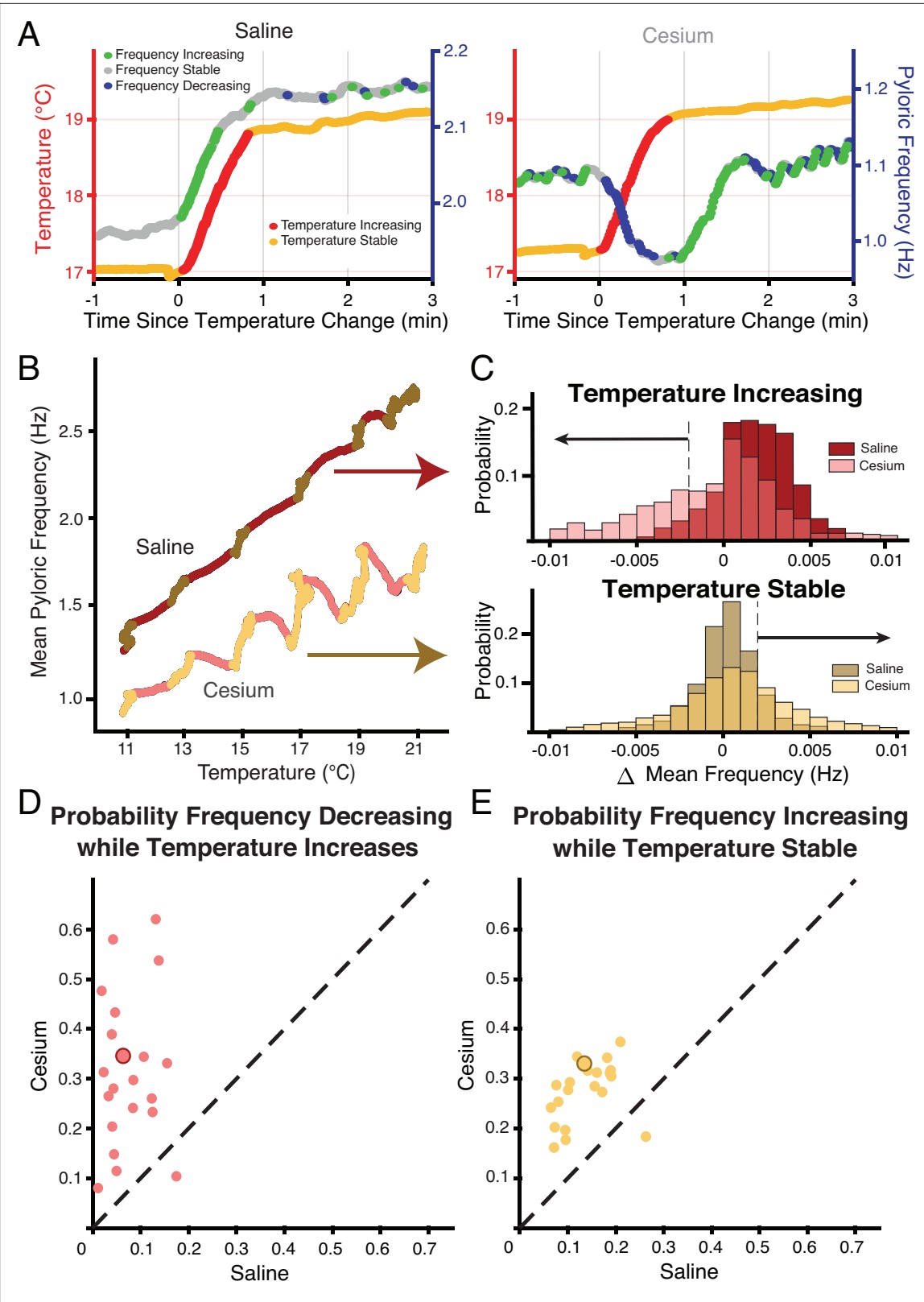

**Figure 3.** Across preparations frequency was more likely to decrease when temperature increased and increase when temperature was stable in Cs+ compared to saline. (**A**) Example of a temperature step from 17°C to 19°C in saline (left) and Cs+ (right). Left axis and warm colors denote temperature, with reds denoting temperature increasing by more than 0.01°C/pyloric cycle, and yellows denoting stable temperature (change less than 0.01°C/pyloric cycle). Right axis and cool colors denote frequency, with green indicating mean frequency increase >0.002 Hz, blue indicating mean frequency

*Figure 3 continued on next page*

*Figure 3 continued*

decrease >0.002 Hz, and gray stable frequency between these bounds. In saline, frequency change closely mirrored temperature change. In Cs$^+$, frequency decreased (blue) when temperature increased (red) and increased (green) when temperature stabilized (yellow). (**B**) Mean pyloric frequency as a function of temperature for a single experiment in both saline and Cs$^+$. Note how in saline the frequency increased largely monotonically along the entire temperature range, increasing most when temperature increased (maroon). Conversely in Cs$^+$ the frequency displayed a non-monotonic, sawtooth-like pattern, decreasing when temperature increased (pink) and increasing when temperature was stable (light yellow). (**C**) For the same experiment as in **B**, distribution of mean change in frequency for cycles when the mean temperature was increasing (top) or when the temperature was stable (bottom), definition of temperature same as in **A**, **B**, *Figure 1A*. When the temperature increased, the change in mean frequency was more likely to decrease in Cs$^+$ than in saline. When temperature was stable, the change in mean frequency was more likely to increase in Cs$^+$ than in saline. Dashed line and black arrows highlight the criteria for decreasing and increasing in frequency used for quantification in **D** and **E**. (**D**) Across experiments, the probability of decreasing mean frequency while temperature increased in Cs$^+$ or in saline. For 20/21 of experiments, the probability was higher in Cs$^+$ than saline. Experiment from **B**, **C** is highlighted by the larger, outlined point. (**E**) Across experiments, the probability of having an increasing frequency while temperature was stable. For 20/21 of experiments, the probability was higher in Cs$^+$ than saline. Again, experiment from **B**, **C** is highlighted by the larger, outlined point.

## Phase constancy with increasing temperature was reduced by Cs$^+$; further advance with increasing temperatures in Cs$^+$

Having established that Cs$^+$ alters the phase relationships of PD and LP, we asked whether the phase constancy observed in control conditions in response to temperature increase was preserved in Cs$^+$. *Figure 6A* illustrates the phase of each pyloric component across temperature for both control and Cs$^+$ conditions. We fit a line to these points for each preparation and condition to determine as a first approximation whether there was a change in phase with temperature, though change in phase with temperature may not have been strictly linear (*Figure 6A*). In control conditions, LP OFF modestly advanced with temperature, with a mean advance of 0.07±0.07 per 10°C (Wilcoxon signed-rank test, N=21, p<0.01). This advance is demonstrated in the two *lvn* traces in *Figure 6B*, which depict one cycle of control activity at 11°C (top) and 21°C (bottom) scaled to the same period. The lilac bar highlights the advance of LP OFF at 21°C. All other pyloric components align, demonstrating phase constancy.

In Cs$^+$, LP ON and LP OFF advanced with temperature, with a mean advance of 0.05±0.05 per 10°C and 0.14±0.09 per 10°C respectively (Wilcoxon signed-rank test, N=21, p<0.01). The advances of LP phase are shown in the two *lvn* traces in *Figure 6C*, which depict one cycle of Cs$^+$ activity at 11°C (top) and 21°C (bottom) scaled to the same period. The purple bar highlights the LP ON advance and the lilac bar highlights the advance of LP OFF at 21°C. PD and PY phase constancy was not significantly altered in either condition. Therefore, LP phase is most sensitive to temperature and is further phase advanced by increased temperatures in Cs$^+$.

## Discussion

Motor patterns operate over a variety of conditions with varying phase and/or frequency to meet environmental needs. While the physiological mechanisms underlying many motor patterns have been investigated under stable conditions, studies of how these mechanisms change across conditions are rarer. We investigated how I$_h$ contributes to frequency and phase of the pyloric rhythm in *C. borealis* during temperature changes using Cs$^+$ to decrease I$_h$. We conclude I$_h$ plays an important role in the ability of the system to appropriately adjust to temperature perturbations both transiently and persistently. Transiently, I$_h$ removal revealed non-monotonic jags in the frequency response mediated by intrinsic properties of individual neurons. Persistently, I$_h$ removal determined the overall frequency change (temperature sensitivity) through network effects.

## The jags in frequency during transitory temperature changes upon I$_h$ removal may be a consequence of imbalances in temperature sensitivities of individual neuron intrinsic properties

We unexpectedly found that in the presence of Cs$^+$, pyloric frequency decreased while temperature increased and then increased after the temperature stabilized (*Figures 2 and 3*), producing jags in the frequency-temperature plot. This dynamic was mediated by the pacemaker alone (*Figure 4*). Previous observations of the effect of temperature on pyloric frequency showed the smooth increase

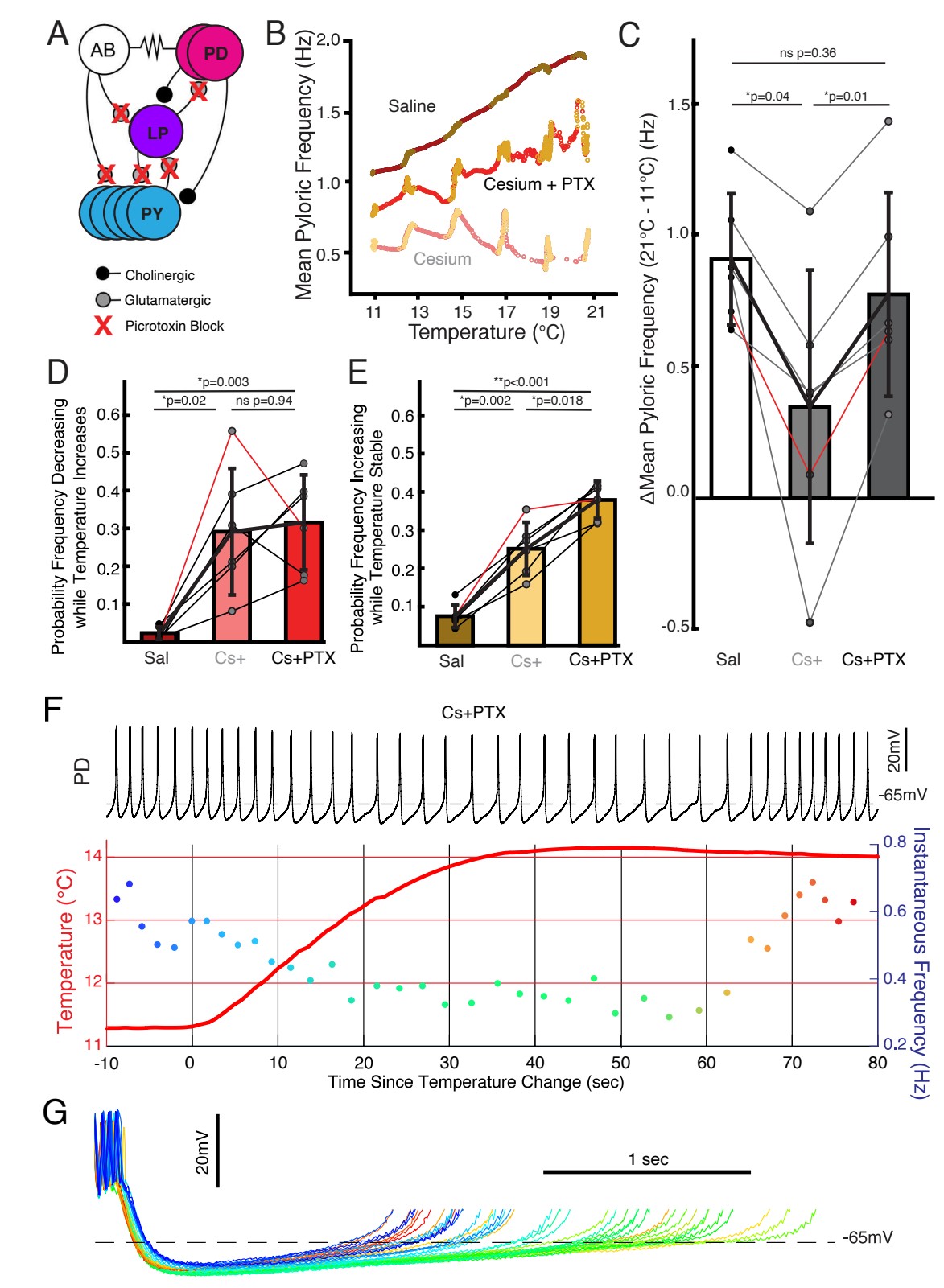

**Figure 4.** Follower neurons mediated pyloric frequency-temperature sensitivity change by Cs+, while non-monotonic temperature response was mediated by the pacemaker kernel neurons. (**A**) Wiring diagram of the pyloric network, illustrating the pacemaker kernel neurons (anterior burster [AB] and pyloric dilator [PD]) and the follower neurons (lateral pyloric [LP] and pyloric [PY]). The only feedback synapse to the pacemaker kernel neurons, LP to PD, is blocked by picrotoxin (PTX). (**B**) Mean pyloric frequency as a function of temperature for a single experiment in saline, Cs+, and Cs++PTX. Red

*Figure 4 continued on next page*

*Figure 4 continued*

dots denote temperature increasing by more than 0.01°C/cycle, yellow dots denote stable temperature (change less than 0.01°C/cycle). In Cs$^+$+PTX, the steady-state increase in frequency (yellows) as a function of temperature was similar to saline (trajectory roughly parallel). However, in Cs$^+$+PTX pyloric frequency displayed non-monotonic responses to periods of transitory temperature increase as in Cs$^+$ alone. (C) There was a significant effect of condition on the change in frequency between 11°C and 21°C for N=6 experiments as a measure of temperature sensitivity (one-way repeated measures ANOVA; $F_{(2,10)}$=12.272, p=0.002). Post hoc Tukey's honestly significant difference (HSD) indicated the frequency changed significantly less in Cs$^+$ alone compared to either control or Cs$^+$+PTX. Experiment from **B** is highlighted in red. (D) There was a significant effect of condition on the probability of frequency decrease during periods of temperature increase for N=6 experiments (one-way repeated measures ANOVA; $F_{(2,10)}$=13.077, p=0.002). HSD indicated the probability was significantly higher in both Cs$^+$ and Cs$^+$+PTX compared to control. Experiment from **B** is highlighted in red. (E) There was a significant effect of condition on the probability of frequency increase during periods of temperature stability for N=6 experiments (one-way repeated measures ANOVA; $F_{(2,10)}$=69.572, p<0.001). HSD indicated the probability was significantly greater in Cs$^+$ than in control and significantly greater in Cs$^+$+PTX than in Cs$^+$. Experiment from **B** is highlighted in red. This elevated probability of increase may partially account for the lack of overall temperature-sensitivity decrease in Cs$^+$+ PTX compared to Cs$^+$ alone in **C**. (F) Same scheme as *Figure 2B and C*. In Cs$^+$+PTX frequency decreased when temperature rose and eventually increased once temperature stabilized. (G) Individual cycles from **F** aligned to the first spike, colored as in the instantaneous frequency trace in **F**. When the temperature increased (greenish traces), the depolarizing phase dramatically elongated and there was some increased hyperpolarization. After approximately 30 s of stable temperature, frequency increased and the depolarization accelerated again (reddish traces).

seen in *Figure 3A* (left) and as such focused on the monotonic increase of steady-state response as in *Figure 1*; (*Tang et al., 2010*; *Rinberg et al., 2013*; *Marder and Rue, 2021*). We highlight for the first time the existence of a more dynamic response to increasing temperature. This dynamic occurred over the ~30 s of temperature increase and the first ~30 s of temperature stability (*Figure 2A*, right, and *Figure 4F*), suggesting the existence of a longer timescale process than previously recognized.

This longer timescale may be the result of interactions between two or more channels and/or channel properties with fast kinetics (*Goaillard et al., 2010*; *Connor, 1975*; *Connor, 1978*; *Connor et al., 1977*). Based on observations that the frequency decrease is associated with an elongation of the depolarizing phase of a burst (*Figures 2C and 4G*), we hypothesize that the decrease in frequency may be primarily caused by effects of temperature on the parameters of the transient potassium current, $I_A$. $I_A$ is an inactivating, hyperpolarizing current activated by depolarization; its activation slows depolarization (*Tang et al., 2010*; *Harris-Warrick et al., 1995*; *MacLean et al., 2005*; *Greenberg and Manor, 2005*; *Tierney and Harris-Warrick, 1992*). Each of its state transitions (activation, inactivation, deinactivation) may be differentially affected by temperature (*Tang et al., 2010*). If deinactivation is more accelerated or altered by temperature than inactivation, then as temperature increases a larger pool of $I_A$ channels would be available to be activated, resulting in transiently more $I_A$ during the depolarizing phase of a PD cycle. This proposed transient increase in $I_A$ would account for the decrease in frequency and elongated depolarization we observed. While temperature continued to change, the difference in parameters would continue to grow. Once the temperature stabilized a new equilibrium between activation, inactivation, and deinactivation could be achieved, potentially resulting in decreased total $I_A$ during a pyloric cycle, an increase in the rate of depolarization, and an increase in frequency. $I_h$ opposes and is correlated with $I_A$ (*Schulz et al., 2007*; *Zhang et al., 2003*; *MacLean et al., 2005*) and prevents a neuron from spending as much time in the $I_A$-deinactivating hyperpolarized state. Thus, when intact, we expect $I_h$ obscures the dynamic frequency effects we propose to be caused by differential temperature sensitivities in $I_A$ parameters.

Mathematically, activity patterns like those seen in Cs$^+$ where the output resulting from a step input first goes 'in the wrong direction' have been described in control theory as having a 'non-minimum phase dynamic' (*Franklin and Powell, 2002*; *Khammash, 2016*). Typically, non-minimum phase systems are fragile and difficult to control, as they are prone to positive feedback loops unless the rate of sensor-induced adjustment is slow. While not uncommon in physics, such dynamics are generally avoided by biological systems to ensure robust responses (*Khammash, 2016*). Our data suggest $I_h$ plays an important role in preventing the system from producing such turbulent behavior, resulting in the typical near-instantaneous and smooth increase in frequency upon changes in temperature. Future modeling may help demonstrate this principle in more general settings.

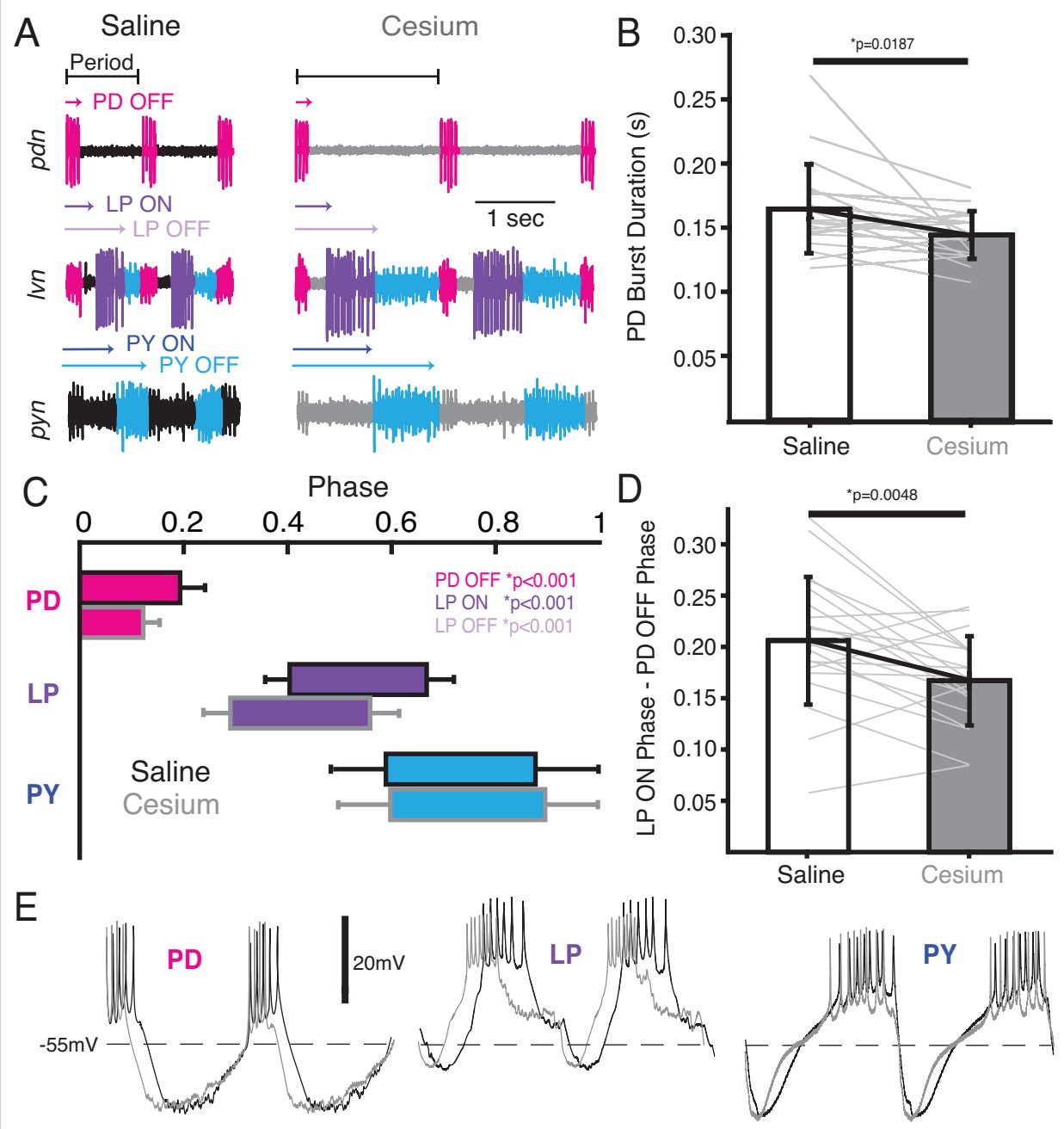

**Figure 5.** In Cs⁺, pyloric dilator (PD) OFF, lateral pyloric (LP) ON, and LP OFF were all advanced compared to saline. (**A**) Extracellular example traces of two pyloric cycles at 11°C recorded simultaneously on the pyloric dilator nerve (*pdn*) (top), lateral ventricular nerve (*lvn*) (middle), and pyloric constrictor nerve (*pyn*) (bottom) in saline (left) and from the same experiment in Cs⁺ (right). PD spikes are colored magenta, LP purple, and pyloric (PY) blue. Onset and offset delay of each neuron relative to the beginning of the period (PD burst) are indicated by arrows. (**B**) Duration of PD burst was marginally but significantly shorter in Cs⁺ than saline (Wilcoxon signed-rank test, N=21, p=0.019). (**C**) Phase plot of saline and Cs⁺ colored as in **A**. PD OFF, LP ON, and LP OFF are all advanced in Cs⁺ compared to saline. (**D**) The phase delay between LP ON and PD OFF is smaller in Cs⁺ than in saline (paired Student's t-test, N=21, p=0.0048). As such, LP ON is more advanced than one would expect if the system was maintaining a constant phase delay between PD OFF and LP ON. (**E**) Two periods of example intracellular recordings from PD (left), LP (middle), and PY (right) in saline (black) and Cs⁺ (gray) scaled to the same period. Each column is from a different experiment. Note the advance of PD OFF, LP ON, and LP OFF with relatively little other change in the overall trajectory of the voltage trace.

**Table 1.** Mean phase of pyloric neurons in saline and Cs$^+$ at 11°C.

Phases that were significantly different in the two conditions are bolded. T=paired Student's t-test, W=Wilcoxon signed-rank test.

| Phase | Saline | Cs$^+$ | Significance test | N | p-Value |
|---|---|---|---|---|---|
| PD OFF | 0.19±0.04 | 0.12±0.03 | T | 21 | <0.001* |
| LP ON | 0.40±0.05 | 0.29±0.05 | T | 21 | <0.001* |
| LP OFF | 0.66±0.05 | 0.56±0.06 | T | 21 | <0.001* |
| PY ON | 0.58±0.11 | 0.60±0.11 | T | 12 | 0.86 |
| PY OFF | 0.87±0.20 | 0.90±0.19 | W | 12 | 0.91 |

## I$_h$ dictates the steady-state increase in pyloric frequency to temperature through network effects

Given that previous studies showed I$_h$ increases in PD and LP cells at elevated temperatures (*Tang et al., 2010*; *Peck et al., 2006*), it is unsurprising that blocking I$_h$ with Cs$^+$ had a larger effect at higher temperatures; the difference in I$_h$ between control and Cs$^+$ is expected to be larger at elevated temperatures. As I$_h$ is a depolarizing current, its removal would be consistent with the larger difference

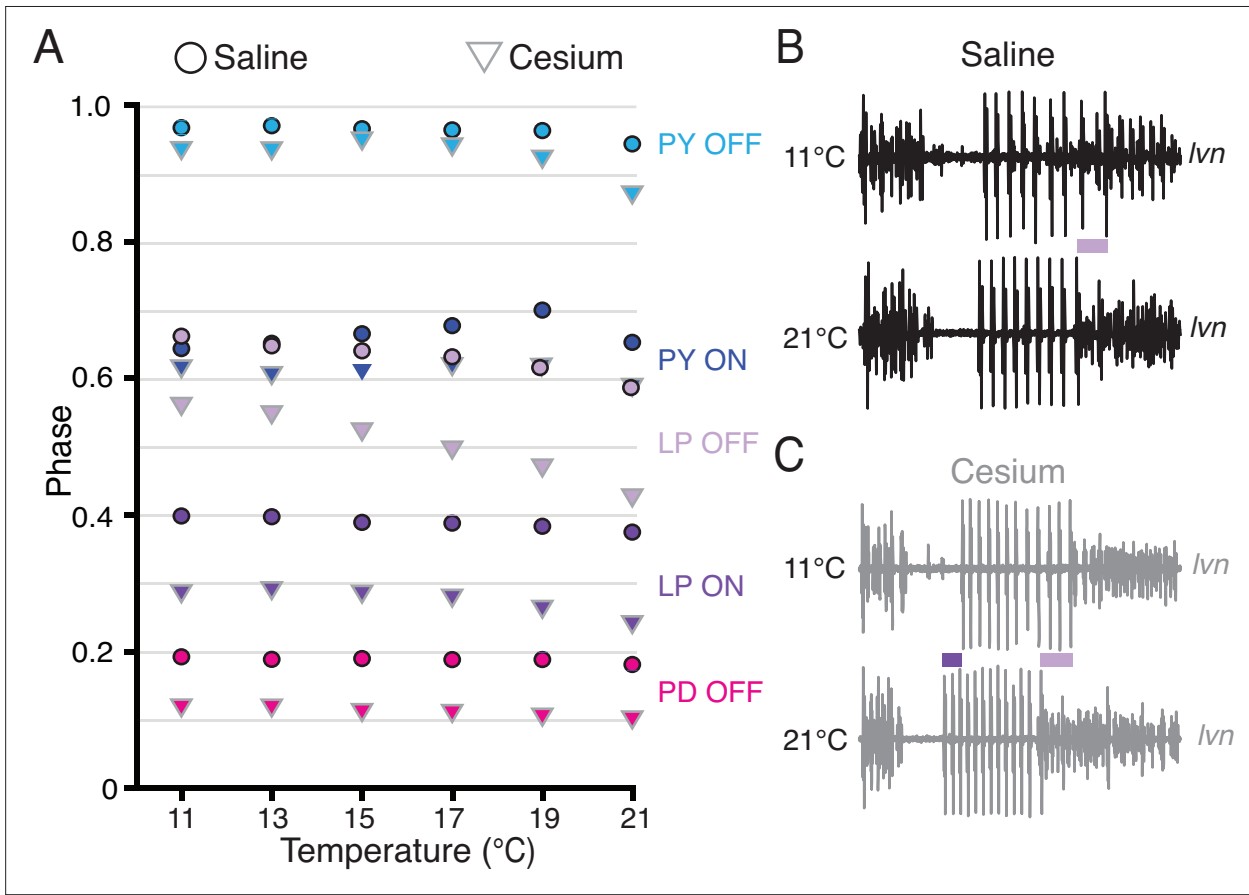

**Figure 6.** Phase advanced across temperatures for lateral pyloric (LP) OFF in both condition; LP ON phase also advanced with increased temperature in Cs$^+$. (**A**) Mean phase as a function of temperature in both saline (circles) and Cs$^+$ (triangles) (N=21 for PD and LP, N=12 for PY). Error bars have been omitted for clarity. LP OFF advanced as temperature increased in both saline and Cs$^+$, though the advance in Cs$^+$ was more dramatic. LP ON also significantly advanced with temperature in Cs$^+$. Criteria for maintaining constancy was a mean slope of phase vs temperature across experiments that was significantly different from zero according to a Wilcoxon signed-rank test (if not-normally distributed) or a paired Student's t-test (if normally distributed), Bonferoni-adjusted. (**B**) Single period of pyloric rhythm recorded extracellularly on the lateral ventricular nerve (*lvn*) at 11°C (top) and 21°C (bottom) scaled to the same period. The advance in LP OFF is highlighted by the lilac bar. (**C**) Same preparation as in **B** but while in Cs$^+$. Lilac bar highlights the LP OFF phase advance, purple bar highlights LP ON advance.

in frequency between saline and Cs$^+$ seen at higher temperatures (*Figure 1D*). Unexpectedly, we found control levels of temperature-induced frequency increase in the presence of Cs$^+$ when the pacemaker kernel was isolated by PTX (*Figure 4B and C*). This result suggests that when temperature increases, the pacemaker kernel can increase the frequency of the pyloric rhythm without the contribution of depolarizing I$_h$. However, if the follower neurons can feed back to the pacemaker and I$_h$ is blocked, the feedback limits the increase in frequency with increased temperature.

Potentially the temperature-induced increase in I$_h$ seen in LP (*Tang et al., 2010*) is partially responsible for permitting the temperature-induced frequency increase seen in control conditions, and its removal by Cs$^+$ decreased the temperature-induced frequency increase. We noted that in Cs$^+$, LP phase advanced with a preserved duty cycle (*Figure 5C*) and advanced further with increased temperature compared to control conditions (*Figure 6A and C*). According to experiments on the effects of differently timed inhibition on pyloric frequency (phase response curve [PRC]), phase advance of LP would be expected to either increase frequency or have no effect (*Mamiya and Nadim, 2004*; *Nadim et al., 2011*; *Weaver and Hooper, 2003*; *Thirumalai et al., 2006*; *Johnson et al., 2011*; *Hooper et al., 2015*), contrary to what we observed. However, temperature-induced I$_h$ increase in the pacemaker kernel (*Peck et al., 2006*) is expected to advance the PRC (*Soofi and Prinz, 2015*), which could permit the increase in frequency with temperature observed in control conditions despite LP inhibition. In Cs$^+$, this advance of the PRC would not occur (*Soofi and Prinz, 2015*), potentially causing the observed reduction in temperature-induced frequency increase in Cs$^+$ (*Figure 1B–E*). In Cs$^+$+PTX, there is no inhibition from LP to potentially slow down frequency, so compensation by increased I$_h$ in PD would be unnecessary. Overall, while I$_h$ is often associated with rhythmogenesis in intrinsic oscillators (*Robinson and Siegelbaum, 2003*; *Tang et al., 2010*; *Brown et al., 1979*), our results demonstrate its increasing role in appropriate regulation of persistent oscillatory frequency though network feedback at elevated temperature.

## LP phase in Cs$^+$

Surprisingly, application of Cs$^+$ phase advanced LP, even more than might be expected when taking into account the advance in PD offset (*Figure 5C and D*). Previous studies reported no significant advance in LP, though these studies were in *Panulirus interruptus* (*Peck et al., 2006*; *Rodgers et al., 2011*). Latency to LP firing increased after Cs$^+$ application but was insufficient to maintain or delay LP's phase given the increased period at 11°C. Increased period is known to advance LP onset (*Mamiya and Nadim, 2004*; *Hooper, 1997*; *Martinez et al., 2019*), but our observed LP onset advance was maintained even in period-matched samples between Cs$^+$ and control conditions. While in these matched samples PD bursting was shorter in Cs$^+$, the time by which PD offset advanced was almost half of what LP advanced and did not significantly correlate with how much LP advanced. The PD to LP synapse is depressing (*Manor et al., 1997*; *Manor et al., 2003*; *Mouser et al., 2008*), such that longer recovery times between PD bursts, such as seen in period-matched Cs$^+$ samples, would be expected to strengthen the synapse, resulting in longer latencies to LP bursting, not shorter. Additionally, PY onset did not advance in period-matched samples. While the synapse from the pacemaker cells to PY and LP are not the same, they have been reported to have similar properties (*Rabbah and Nadim, 2005*) and there is no reason a priori to believe that Cs$^+$ differentially affected these synapses.

It was also surprising that when I$_h$ was blocked, LP firing *further* advanced with temperature-mediated frequency increases. Phase maintenance is sometimes explained through a balance of synaptic depression and I$_A$ (*Greenberg and Manor, 2005*; *Manor et al., 2003*; *Mouser et al., 2008*; *Bose et al., 2004*). The models of exploring phase constancy in oscillatory networks reminiscent of the STG (*Greenberg and Manor, 2005*; *Manor et al., 2003*) suggest that given I$_h$ acts in opposition to I$_A$, normally I$_h$ should impair phase constancy by *shortening* LP latencies as the period increases and advancing LP, and that this effect should be minimized when period is decreasing, such as was seen when temperature increased. By this logic, I$_h$ removal would be expected to improve phase constancy and to have less effect on phase as period decreased, which is the opposite of what we observed. It therefore remains unclear why in Cs$^+$ LP was advanced and then further advanced as frequency increased with elevated temperatures. Again, more detailed modeling in the future may shed light on the precise property interactions that produce this unexpected further phase advancement.

While Cs$^+$ has been reported to block channels other than those responsible for I$_h$ (*Thoby-Brisson et al., 2000*; *Pape, 1996*; *Wischmeyer and Karschin, 1997*; *Coggan et al., 1994*; *Harding et al.,*

*2024*; *Brown and Adams, 1980*), there is no evidence that these other channels are playing significant roles in the current preparation (*Zhu et al., 2016*). Moreover, Cs⁺ primarily blocks potassium-passing channels in a voltage-dependent manner (*Clay and Shlesinger, 1984*) such that only inward currents are blocked when Cs⁺ is extracellular. Additionally, we did not observe an increase in pyloric frequency or a consistent intracellular depolarization in Cs⁺, both of which would be expected if Cs⁺ were indirectly changing the extracellular potassium concentration through effects on other potassium channels. Therefore, as a first approximation most of the effects seen here can likely be attributed to Cs⁺ block of $I_h$.

## Concluding remarks and general implications

There was substantial animal to animal variability in the effect of Cs⁺ block of $I_h$. Likely this variability stems from the degeneracy in the combinations of currents that can result in a particular motor pattern (*Alonso and Marder, 2019*; *Schulz et al., 2006*; *Prinz et al., 2004*; *Krenz et al., 2015*; *Goaillard et al., 2009*; *Anwar et al., 2022*; *Marder et al., 2022*). Some combinations likely rely more on $I_h$ to produce the prototypical response to temperature. Despite this variability, without $I_h$, the pyloric network was unable to reliably produce either its typical rapid and smooth increase in frequency in response to temperature change, or its typical steady-state increase in frequency. As such $I_h$ may act as a buffer to transient response hiccups that may be caused by intrinsic properties and as a buffer to the limiting effects of network inhibition.

## Methods

### Animals and experimental preparation

*C. borealis* were purchased from Commercial Lobster (Boston, MA, USA) and maintained in tanks containing artificial seawater (Instant Ocean) at 11–13°C on a 12 hr light/dark cycle for at least 1 week before use. Estimated age was ~7 years. Crabs were acquired between November 2022 and December 2023. All experiments are biological replicates. After placing the animals on ice for 30 min, the stomatogastric nervous system was dissected from the crab and pinned in a Petri dish coated with Sylgard (Dow Corning) as described previously (*Gutierrez and Grashow, 2009*). All preparations included the STG, the eosophageal ganglia, and two commissural ganglia. The STGs were desheathed to allow a combination of intracellular recordings and direct solution contact with the STG. The nervous system was kept in physiological saline composed of 440 mM NaCl, 11 mM KCl, 13 mM CaCl₂, 26 mM MgCl₂, 11 mM Trizma base, and 5 mM maleic acid, pH 7.4–7.5 at 23°C (~7.7–7.8 pH at 11°C). All reagents were purchased from Sigma.

### Temperature protocol

Temperature of superfused saline solution was controlled using an SC-20 Peltier device controlled by model CL-100 temperature controller (Warner Instruments). Temperature was monitored with a Warner Instruments TA-29 thermocouple placed within 1 cm of the STG. During 'Step' protocols, temperature was set manually to within 0.2°C of 11°C. After 2 min of recording activity, the temperature was manually increased by 2°C and allowed to stabilize over 4 min. This process was repeated until the dish reached 21°C, after which the temperature was returned to 11°C (*Figure 1A*). The preparation was kept at 11°C for 10 min during which the solution was changed. Experiments started with either standard physiological saline or saline with 5 mM CsCl obtained from Sigma to block $I_h$ (*Peck et al., 2006*; *Harris-Warrick et al., 1995*; *Goeritz et al., 2011*; *Ballo et al., 2010*). Experiments to test the importance of follower neurons in the various effects of Cs⁺ were performed with first randomized Cs⁺ or saline temperature steps protocol, followed by a temperature step protocol in Cs⁺+10⁻⁵ M picrotoxin (PTX [*Bidaut, 1980*]) from Sigma as PTX does not easily wash out.

### Electrophysiology

Extracellular recordings were made by placing wells around the lower *lvn*, *pdn*, and *pyn*. Wells were made from 90% Vaseline 10% mineral oil solution. Stainless-steel pin electrodes were placed within the wells to monitor spiking activity and amplified using Model 1700 Differential AC Amplifiers (A-M Systems). Intracellular recordings from the soma of the PD, LP, and PY neurons were made using either discontinuous current clamp (one-electrode) or two-electrode current and/or voltage clamp with

5–20 MΩ sharp glass microelectrodes filled with 0.6 M $K_2SO_4$ and 20 mM KCl solution. For voltage clamp experiments, 3 M KCl solutions were used to facilitate sustained current injections. Intracellular signals were amplified with an Axoclamp 900 A amplifier (Molecular Devices, San Jose, CA, USA). Cells were identified by matching intracellular activity with activity on the *pyn* (PY cells), *pdn* (PD cells), or *lvn* (LP cells). All amplified signals were digitized using a Digidata 1440 digitizer (Molecular Devices, San Jose, CA, USA) and recorded using pClamp data acquisition software (Molecular Devices, San Jose, CA, USA, version 10.5), sampling frequency of 10 kHz.

## Analysis

Spikes were detected and assigned to a particular neuron using a custom software implemented in MATLAB ('Crabsort', https://github.com/sg-s/crabsort, *Gorur-Shandilya, 2022*; *Powell et al., 2021*; *Gorur-Shandilya et al., 2022*) which used a machine-learning algorithm. Results of this automated process were double-checked and edited by hand. Bursts were identified as two or more consecutive PD spikes with an inter-burst interval of at least 200 ms. Phase was defined as the difference between the first spike of a PD burst and the first (onset) or last (offset) spike of a neuron (LP, PY, or PD), normalized by the period (time between first PD spikes of two consecutive bursts). Data were tested for normality using the Shapiro-Wilk test and appropriate parametric or non-parametric tests were used.

To quantify the non-monotonic responses to temperature change, we first smoothed frequency and temperature using a moving average over 30 and 10 pyloric cycles respectively to get a less noisy trajectory for each measure. We then divided pyloric cycles based on whether the smoothed temperature increased more than 0.01°C. For cycles that increased more than this criterion, we quantified the percent in which the smoothed frequency decreased more than 0.002 Hz per cycle. For cycles where temperature did not increase above the criterion, we quantified the percent in which the smoothed frequency increased more than 0.002 Hz per cycle.

To examine phase constancy, we found the best fit line to describe the relationship between mean phase and temperature for each experiment. We then used either Wilcoxon signed-rank test (non-normal data) or one-sample t-test (normal data) to determine whether the slope across experiments was significantly different from zero. Due to the 10 different tests (5 phases × 2 conditions), we Bonferroni-adjusted the p-values to reduce the false-positive rate.

Where not specifically noted otherwise, all summary statistic reports are (mean ± SD). All analysis was performed using MATLAB.

## Acknowledgements

We thank Maria Ivanova and Kathleen Jacquerie for their advice on early drafts of the manuscript. We thank Thiago Burghi, Leandro Alonso, and Kathleen Jacquerie, for their helpful discussions on dynamical systems. We would also like to thank Sonal Kedia and Margret Lee for their aid in early setup of experimental equipment.

## Additional information

### Funding

| Funder | Grant reference number | Author |
| --- | --- | --- |
| National Institute of Neurological Disorders and Stroke | T32NS007292 | Kyra Schapiro |
| National Institute of Neurological Disorders and Stroke | R35NS097343 | Eve Marder |

The funders had no role in study design, data collection and interpretation, or the decision to submit the work for publication.

## Author contributions
Kyra Schapiro, Conceptualization, Formal analysis, Investigation, Visualization, Writing - original draft, Writing – review and editing; JD Rittenberg, Formal analysis, Investigation; Max Kenngott, Investigation; Eve Marder, Conceptualization, Supervision, Funding acquisition, Writing – review and editing

## Author ORCIDs
Kyra Schapiro ⓘ https://orcid.org/0000-0001-8308-0744
JD Rittenberg ⓘ http://orcid.org/0009-0008-5742-6340
Max Kenngott ⓘ http://orcid.org/0000-0002-3452-6508
Eve Marder ⓘ https://orcid.org/0000-0001-9632-5448

Reviewer #1 (Public review): https://doi.org/10.7554/eLife.98844.3.sa1
Reviewer #2 (Public review): https://doi.org/10.7554/eLife.98844.3.sa2
Reviewer #3 (Public review): https://doi.org/10.7554/eLife.98844.3.sa3
Author response https://doi.org/10.7554/eLife.98844.3.sa4

## Additional files

### Supplementary files
• MDAR checklist

### Data availability
Code and data to reproduce the results and figures in this work was uploaded to Dryad: https://doi.org/10.5061/dryad.1rn8pk129.

The following dataset was generated:

| Author(s) | Year | Dataset title | Dataset URL | Database and Identifier |
|---|---|---|---|---|
| Schapiro KA, Rittenberg JD, Kenngott M, Marder E | 2024 | Effects of Cs on temperature response of *C. borealis* pyloric rhythm | https://doi.org/10.5061/dryad.1rn8pk129 | Dryad Digital Repository, 10.5061/dryad.1rn8pk129 |

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
