## [Editor Report · eLife assessment]

This **important** study investigates neurobiological mechanisms underlying the maintenance of stable, functionally appropriate rhythmic motor patterns during changing environmental conditions - temperature in this study in the crab *Cancer borealis* stomatogastric central neural pattern generating circuits producing the rhythmic pyloric motor pattern, which is naturally subjected to temperature perturbations over a substantial range. The authors present **compelling** evidence that the neuronal hyperpolarization-activated inward current (Ih), known to contribute to rhythm control, plays a vital role in the ability of these circuits to appropriately adjust the frequency of rhythmic neural activity in a smooth monotonic fashion while maintaining the relative timing of different phases of the activity pattern that determines proper functional motor coordination transiently and persistently to temperature perturbations. This study will be of interest to neurobiologists studying rhythmic motor circuits and systems and their physiological adaptations.

---

## [Referee Report · Reviewer #1 (Public review)]

Summary:

This important study investigates the neurobiological mechanisms underlying the stable operation and maintenance of functionally appropriate rhythmic motor patterns during changing environmental conditions - temperature in this study in the crab *Cancer borealis* stomatogastric neural pattern generating network producing the pyloric motor rhythm, which is naturally subjected to temperature perturbations over a substantial range. This study is relevant to the general problem that some rhythmic motor systems adjust to changing environmental conditions and state changes by increasing the cycle frequency in a smooth monotonic fashion while maintaining the relative timing of different network activity pattern phases that determine proper motor coordination. How this is achieved mechanistically in complex dynamic motor networks is not understood, particularly how the frequency and phase adjustments are achieved as conditions change while avoiding operational instabilities on different time scales. The authors specifically studied the contributions of the hyperpolarization-activated inward current (Ih), which is involved in rhythm control, to the adjustments of frequency and phases in the pyloric rhythmic pattern as the temperature was altered from 11 degrees C to 21 degrees C. They present compelling evidence that this current is a critical biophysical feature in the ability of this system to adjust transiently and persistently to temperature perturbations appropriately. After blocking Ih in the pyloric network with cesium, the network was unable to reliably produce its characteristic rapid and smooth increase in the frequency of the triphasic rhythmic motor pattern in response to increasing temperature or its typical steady-state increase in frequency over this Q10 temperature range.

Strengths:

(1) The authors addressed this problem by technically rigorous experiments in the crab *Cancer borealis* stomatogastric ganglion (STG) in vitro, which readily allows for neuronal activity recording in a behaviorally and architecturally defined rhythmic neural circuit in conjunction with the application of blockers of Ih and synaptic receptors to disrupt circuit interactions. This approach is an effective way to experimentally investigate how complex rhythmic networks, at least in poikilotherms, mechanistically adjust to environmental perturbations such as temperature.

(2) While previous work demonstrated that Ih increases in pyloric neurons as temperature increases, the authors here establish that this increase is necessary for normal responses of STG neural activity to temperature, which consist of a smooth monotonic increase in the frequency of rhythmic activity with increasing temperature.

(3) The data shows that blocking Ih with cesium causes the frequency to transiently decrease ("jags") when the temperature increases and then increases after the temperature stabilizes at a steady state, revealing a non-monotonic frequency response to temperature perturbations.

(4) The authors dissect some of the underlying neuronal and circuit dynamics, presenting evidence that after blocking Ih, the non-monotonic jags in the frequency response are mediated by intrinsic properties of pacemaker neurons, while in the steady state, Ih determined the overall frequency change (i.e., temperature sensitivity) through network interactions.

(5) The authors' results highlight more complex dynamic responses to increasing temperature for the first time, suggesting a longer timescale process than previously recognized that may result from interactions between multiple channels and/or ion channel kinetics.

Weaknesses:

(1) The involvement of Ih in achieving the frequency and phase adjustments as conditions change and allowing smooth transitions to avoid operational instabilities in other complex rhythmic motor networks, for example, in homeotherms, is not established, so the present results may have limited general extrapolations.

---

## [Referee Report · Reviewer #2 (Public review)]

Summary:

Using the crustacean stomatogastric nervous system (STNS), the authors present an interesting study wherein the contribution of the Ih current to temperature-induced changes in the frequency of a rhythmically active neural circuit is evaluated. Ih is a hyperpolarization-activated cation current that depolarizes neurons. Under normal conditions, increasing the temperature of the STNS increases the frequency of the spontaneously active pyloric rhythm. Notably, under normal conditions, as temperature systematically increases, the concomitant increase in pyloric frequency is smooth (i.e., monotonic). By contrast, blocking Ih with extracellular cesium produces temperature-induced pyloric frequency changes that follow a characteristic sawtooth response (i.e., non-monotonic). That is, in cesium, increasing temperature initially results in a transient drop in pyloric frequency that then stabilizes at a higher frequency. Thus, the authors conclude that Ih establishes a mechanism that ensures smooth changes in neural network frequency during environmental disturbances, a feature that likely bestows advantages to the animal's function.

The study describes several surprising and interesting findings. In general, the study's primary observation of the cesium-induced sawtooth response is remarkable. To my knowledge, this type of response has not yet been described in neurobiological systems, and I suspect that the unexpected response will be of interest to many readers.

At first glance, I had some concerns regarding the use of extracellular cesium to understand network phenomena. Yes, extracellular cesium blocks Ih. But extracellular cesium has also been shown to block astrocytic potassium channels, at least in mammalian systems (i.e., K-IR, PMID: 10601465), and such a blockade can elevate extracellular potassium. I was heartened to see that the authors acknowledge the non-specificity of cesium (lines 320-325) and I agree with the authors' contention that "a first approximation most of the effects seen here can likely be attributed to Cs+ block of Ih". Upon reflecting on the potential confound, I was also reassured to see that extracellular cesium alone does not in fact increase pyloric frequency, an effect that might be expected if cesium indirectly raises [K^+^]outside. If the authors agree, then I suggest including that point in their discussion.

In summary, the authors present a solid investigation of a surprising biological phenomenon. In general, my comments are fairly minor. Thanks for contributing an interesting study.

Strengths:

A major strength of the study is the identification of an ionic conductance that mediates stable, monotonic changes in oscillatory frequency that accompany changes in the environment (i.e., temperature).

Weaknesses:

A potential experimental concern stems from the use of extracellular cesium to attribute network effects specifically to Ih. Previous work has shown that extracellular cesium also blocks inward-rectifier potassium channels expressed by astrocytes, and that such blockade may also elevate extracellular potassium, an action that generally depolarizes neurons. Notably, the authors address this potential concern in the discussion.

---

## [Referee Report · Reviewer #3 (Public review)]

Summary:

This paper presents a systematic analylsis of the role of the hyperpolarization-activated inward current (the h current) in the response of the pyloric rhythm of the stomatogastric ganglion (STG) of the crab. In a detailed set of experiments, they analyze the effect of blocking h current with bath infusion of the h current blocker cesium (perfused as CsCl). They show interesting and reproducible effects that blockade of h current results in a period of frequency decrease after an upward step in temperature, followed by a slow increase in frequency. This contrasts with the normal temperature response that shows an increase in frequency with an increase in temperature without a downward "jag" in the frequency response. This is an important paper for showing the role of h current in stabilizing network dynamics in response to perturbations such as a temperature change.

Strengths of the paper:

The major effects are shown very clearly and convincingly in a range of experiments with combined intracellular recording from neurons during changes in temperature.

Weaknesses

The Marder lab has detailed models of the pyloric rhythm. These temperature effects have not yet been modeled and could be the focus of future modeling studies.

---

## [Author Response]

The following is the authors’ response to the original reviews.

**Response to Public Reviews:**

We thank the reviewers for their kind comments have implemented many of the suggestion their suggestions. Our paper has greatly benefited from their advice. Like Reviewer 1, we acknowledge that while the exact involvement of Ih in allowing smooth transitions is likely not universal across all systems, our demonstration of the ways in which such currents can affect the dynamics of the response of complex rhythmic motor networks provides valuable insight. To address the concerns of Reviewer 2, we included a sentence in the discussion to highlight the fact that cesium neither increased the pyloric frequency nor caused consistent depolarization in intracellular recordings. We also highlighted that these observations suggest both that cesium is not indirectly raising [K+]outside and support the conclusion that the effects of cesium are primarily through blockade of Ih rather than other potassium channels.

Reviewer 3 raised some important points about modeling. While the lab has models that explore the effects of temperature on artificial triphasic rhythms, these models do not account for all the biophysical nuances of the full biological system. We have limited data about the exact nature of temperature-induced parameter changes and the extent to which these changes are mediated by intrinsic effects of temperature on protein structure versus protein interactions/modification by processes such as phosphorylation. With respects to the A current, Tang et al., 2010 reported that the activation and inactivation rates are differentially temperature sensitive but we do not have the data to suggest whether or not the time courses of such sensitivities are different. As such, we focus our discussion on the properties we know are modulated by temperature, i.e. activation rates. Within the discussion we now include the suggestion that future, more comprehensive modeling may be appropriate to further elucidate the ways in which reducing Ih may produce the here reported experimentally observed effects.

**Reviewer #1 (Recommendations For The Authors):**
Suggested revisions:A figure showing examples of the voltage-clamp traces for the critical measurements of the extent of Ih block by 5 mM CsCl in PD and LP neurons at the temperature extremes in these preparations is not shown, and the authors should consider including such a figure, perhaps as a supplemental figure.

We have added Supplemental Figure 1 containing voltage-clamp traces demonstrating the extent of Ih block by 5mM CsCl in PD and LP neurons at 11 and 21°C. Due to technical concerns, different preparations were used in the measurements at 11°C and 21°C, but the point that the H-current is reduced is demonstrated in all cases.

**Reviewer #2 (Recommendations for The Authors):**
Specific (Minor) Comments:(1) Line 83: In Cs+ "at 11°C, the pyloric frequency was significantly decreased compared to control conditions (Saline: 1.2± 0.2 Hz; Cs+ 0.9± 0.2 Hz)".As above, the authors often report that cesium generally reduces pyloric frequency. Figure 5A demonstrates this action quite nicely. However, cesium's effect on pyloric frequency at 11°C seems less robust in Figure 1C. Why the discrepancy?

There is variability in the effects of Cs+ on the pyloric frequency. As noted, the standard deviation in frequency in both conditions is 0.2Hz. As such, there are some cases in which the initial frequency drop in Cs+ compared to control was relatively small. 1C is one such case, but was selected as an example because of its clear reduction in temperature sensitivity.

(2) I don't understand what the arrows/dashed lines are trying to convey in Figure 3C.

The arrows/dashed lines represent the criteria used to define a cycle as “decreasing in frequency” (Temperature Increasing) or “increasing in frequency” (Temperature Stable). We have amended lines 130 and 137 in the text to hopefully clarify this point, as well as the figure legend.

(3) Lines 118/168. The description of cesium's specific action on the depolarizing portion of PD activity is a bit confusing. In my mind, "depolarization phase" refers to the point at which PD is most depolarized. Perhaps restating the phrase to "elongation of the depolarizing trajectory" is less confusing. The authors may also want to consider labeling this trajectory in Figure 2C.

We have changed “depolarization phase” to “depolarizing phase” to highlight that this is the period during which the cell is depolarizing, rather than at its most depolarized. We consider the plateau of the slow wave and spiking (the point at which PD is most depolarized) to be the “bursting phase”. We have labeled these phases in Figure 2C as suggested.

(4) Figure 3C legend: a few words seem to be missing. I suggest "the change in mean frequency was more likely TO decrease IN Cs+ than in saline".

Thank you for catching this typo, it has been corrected.

(5) Line 165: Awkward phrasing. “In one experiment, the decrease in frequency while temperature increased and subsequent increase in frequency after temperature stabilized was particularly apparent in Cs+ PTX”.How about: “One Cs+ PTX experiment wherein elevating the temperature transiently decreased pyloric frequency is shown in Figure 4F.”

We have amended this sentence to read, “One Cs++PTX experiment in which elevating the temperature produced a particularly pronounced transient decrease in frequency is shown in Figure 4F.”

(6) Line 186: Awkward phrasing. "LP OFF was also significantly advanced in Cs+, although duty cycle (percent of the period a neuron is firing) was preserved".The use of the word "although" seems a bit strange. If both LP onset and LP offset phase advance by the same amount, then isn't an unchanged duty cycle expected?

“Although” has been changed to “and subsequently”.

**Reviewer #3 (Recommendations For The Authors):**
Major comments:(1) I know the Marder lab has detailed models of the pyloric rhythm. I am not saying they have to add modeling to this already extensive and detailed paper, but it would be useful to know how much of these temperature effects have been modeled, for example in the following locations.(2) Line 259 - "Mathematically..." - Is there a computational model of H current that has shown this decrease in frequency in pyloric neurons? If you are working on one for the future, you could mention this.

There is not currently a model in which the reduction of the H-current results in the non-minimum phase dynamics in the frequency response to temperature seen experimentally. It should be noted that our existing models of pyloric activity responses to temperature are not well suited to investigate such dynamics in their current iterations. Further work is necessary to demonstrate the principles observed experimentally in computational modeling, and we have added a sentence to the paper to reflect this point (Line 268).

(3) Line 318 - "therefore it remains unclear" - I thought they had models of the circuit rhythmicity. Do these models include temperature effects? Can they comment on whether their models of the circuit show an opposite effect to what they see in the experiment? I'm not saying they have to model these new effects as that is probably an entirely different paper, but it would be interesting to know whether current models show a different effect.

We have some models of the pyloric response to temperature, but these models were specifically selected to maintain phase across the range of temperature. When Ih was reduced in these models, a variety of effects on phase and duty cycle were seen. These models were selected to have the same key features of behavior as the pyloric rhythm, but do not capture all the biophysical nuances of the complete system, and therefore should not necessarily be expected to reflect the experimental findings in their current iterations. Furthermore, these models are meant to have temperature as a static, rather than dynamic input, and thus are ill-suited to examine the conditions of our experiments. The models in their current state are not sufficiently relevant to these experimental findings that we they can illuminate the present paper `2.

(4) "If deinactivation is more accelerated or altered by temperature than inactivation...While temperature continued to change, the difference in parameters would continue to grow" - This is described as a difference in temperature sensitivity, but it seems like it is also a function of the time course of the response to change in temperature (i.e. the different components could have the same final effect of temperature but show a different time course of the change).

We know from Tang et al, 2010, that activation and inactivation rates of the A current are differentially temperature sensitive. We have no evidence to suggest that the time course of the response to temperature of various parameters differ. The physical actions of temperature on proteins are likely to be extremely rapid, making a time course difference on the order of tens of seconds less unlikely, though not impossible. Modeling of the biophysics might illuminate the relative plausibility of these different mechanisms of action, but we feel that our current suggested explanation is reasonable based on existing information.

(5) Is it known how temperature is altering these channel kinetics? Is it via an intrinsic rearrangement of the protein structure, or is it a process that involves phosphorylation (that could explain differences in time course?). Some mention of the mechanism of temperature changes would be useful to readers outside this field.

It is not known exactly how temperature alters channel parameters. Invariably some, if not all, of it is due to an intrinsic rearrangement of protein structure, and our current models treat all parameter changes as an instantaneous consequence. However, it is possible that some effects of temperature are due to longer timescale processes such as phosphorylation or cAMP interactions. Current work in the lab is actively exploring these questions, but there is no definitive answer. Given that this paper focuses on the phenomenon and plausible biomolecular explanations based on existing data, we have not altered the paper to include more exhaustive coverage of all the possible avenues by which temperature may alter channel properties.

Specific comments:Title: misspelling of "Cancer" ?

We are unsure how that extra “w” got into the earliest version of the manuscript and have removed it.

Line 66 "We used 5mM CsCl" - might mention right up front that this was a bath application of the substance.

We have altered this line to read “used bath application of 5mM CsCl”.

Figure 4 - "The only feedback synapse to the pacemaker kernel neurons, LP to PD, and is blocked by picrotoxin" - I think the word "and" should be removed from this phrase in the figure legend.

Fixed

Figure 4 legend - "Reds denote temperature...yellows denote..." - I think it should be "Red dots denote temperature...yellow dots denote...".

Done

Figure 4B - Why does the change in frequency in cesium look so different in Figure 4B compared to Figure 1C or Figure 3B? In the earlier figures, the increase of frequency is smaller but still present in cesium, whereas, in Figure 4B, cesium seems to completely block the increase in frequency. I'm not sure why this is different, but I guess it's because 3B and 4B are just mean traces from single experiments. Presumably, 4B is showing an experiment in which the cesium was subsequently combined with picrotoxin?

Figures 1C, 3B, and 4B are indeed all from different single experiments. As acknowledged in our concluding paragraph, there was substantial variability in the exact response of the pyloric rhythm to temperature while in cesium. The most consistent effect was that the difference in frequency between cesium and saline at a particular temperature increased, as demonstrated across 21 preparations in Figure 1D. It may be noted in Figure 1E that the Q10 was not infrequently <1, meaning that there was a net decrease in frequency as temperature increased in some experiments such as seen in the example of Figure 4B. The “fold over” (initial increase in steady-state frequency with temperature, then decrease at higher temperatures) has been observed at higher temperatures (typically around 23-30 degrees C) even under control conditions but has not been highlighted in previous publications. The example in 4B was chosen because it demonstrated both the similarity in jags between Cs+ and Cs++PTX and an overall decrease in temperature sensitivity, even though in this instance the steady-state change in frequency with temperature was not monotonic.

Figure 6A - "Phase 0 to 1.0" - The y-axis should provide units of phase. Presumably, these are units of radians so 1.0=2*pi radians (or 360 degrees, but probably best to avoid using degrees of phase due to confusion with degrees of temperature).

Phase, with respect to pyloric rhythm cycles, does not traditionally have units as it is a proportion rather than an angle. As such, we have not changed the figure.

Line 275 - "the pacemaker neuron can increase" - Does this indicate that the main effects of H current are in the follower neurons (i.e. LP and PY versus the driver neuron PD)?

Not necessarily. We posit in the next paragraph that the effect of the H current on the temperature sensitivity could be due to its phase advance of LP, but that phase advance of LP is not particularly expected to increase frequency. We favor the possibility that temperature increases Ih in the pacemaker, which in turn advances the PRC of the rhythm, allowing the frequency increase seen under normal conditions. In Cs+, this advance does not occur, resulting in the lower temperature sensitivity. In Cs++PTX, the lack of inhibition from LP means compensatory advance of the pacemaker PRC by Ih is unnecessary to allow increased frequency.

Line 285 - "either increase frequency have no effect" - Is there a missing "or" in this phrase?

Thank you, we have added the “or”.